# Potential Combination Drug Therapy to Prevent Redox Stress and Mitophagy Dysregulation in Retinal Müller Cells under High Glucose Conditions: Implications for Diabetic Retinopathy

**DOI:** 10.3390/diseases9040091

**Published:** 2021-12-14

**Authors:** Lalit Pukhrambam Singh, Takhellambam S. Devi

**Affiliations:** Department of Ophthalmology, Visual and Anatomical Sciences, Wayne State University School of Medicine, Detroit, MI 48201, USA; au6363@wayne.edu

**Keywords:** high glucose, TXNIP, redox stress, mitophagy, retinal Müller cell, diabetic retinopathy

## Abstract

Chronic hyperglycemia-induced thioredoxin-interacting protein (TXNIP) expression, associated oxidative/nitrosative stress (ROS/RNS), and mitochondrial dysfunction play critical roles in the etiology of diabetic retinopathy (DR). However, there is no effective drug treatment to prevent or slow down the progression of DR. The purpose of this study is to examine if a combination drug treatment targeting TXNIP and the mitochondria-lysosome pathway prevents high glucose-induced mitochondrial stress and mitophagic flux in retinal Müller glial cells in culture, relevant to DR. We show that diabetes induces TXNIP expression, redox stress, and Müller glia activation (gliosis) in rat retinas when compared to non-diabetic rat retinas. Furthermore, high glucose (HG, 25 mM versus low glucose, LG 5.5 mM) also induces TXNIP expression and mitochondrial stress in a rat retinal Müller cell line, rMC1, in in vitro cultures. Additionally, we develop a mitochondria-targeted mCherry and EGFP probe tagged with two tandem COX8a mitochondrial target sequences (adenovirus-CMV-2×mt8a-CG) to examine mitophagic flux in rMC1. A triple drug combination treatment was applied using TXNIP-IN1 (which inhibits TXNIP interaction with thioredoxin), Mito-Tempo (mitochondrial anti-oxidant), and ML-SA1 (lysosome targeted activator of transient calcium channel MCOLN1/TRPML1 and of transcription factor TFEB) to study the mitochondrial–lysosomal axis dysregulation. We found that HG induces TXNIP expression, redox stress, and mitophagic flux in rMC1 versus LG. Treatment with the triple drug combination prevents mitophagic flux and restores transcription factor TFEB and PGC1α nuclear localization under HG, which is critical for lysosome biosynthesis and mitogenesis, respectively. Our results demonstrate that 2×mt8a-CG is a suitable probe for monitoring mitophagic flux, both in live and fixed cells in in vitro experiments, which may also be applicable to in vivo animal studies, and that the triple drug combination treatment has the potential for preventing retinal injury and disease progression in diabetes.

## 1. Introduction

Diabetic retinopathy (DR) is a devastating complication of the eyes affecting millions of people around the world. However, no effective therapy to prevent or treat the disease is available so far. Therefore, there is a need for developing new and innovative drug therapies to prevent or slow down the progression of DR. Hence, the purpose of this study is to explore such opportunities through targeting multiple cellular signaling pathways that are affected in DR. The retina is a complex tissue responsible for visual perception and a window to the outside environment [1,2,3,4]. It is composed of three neuronal layers consisting of rod and cone photoreceptors (PRs) in the outer nuclear layer (ONL), bipolar neurons in the inner nuclear layer (INL), and retinal ganglion layer (RGC) that is connected by the outer and inner plexiform layers (OPL and IPL, respectively) (Figure 1). In addition, amacrine and horizontal neurons are also present. Müller cells (MC) expand the entire neuroretina with their end-feet forming the inner limiting membrane (ILM) in the vitreous and outer limiting membrane (OLM) in the photoreceptor, while cell bodies reside in the INL.

Müller cell processes are in contact with the retinal neurovascular system and play important roles in maintaining retinal homeostasis [5]. MCs take up glutamate and recycle glutamine via glutamine synthetase (GS) enzyme to neurons while producing neurotrophic growth factors for neuronal survival. MCs supply glucose and metabolites to neurons as well, and maintain retinal water content and osmolality by inserting aquaporin 4 on the blood vessel, pumping out excess metabolic water [6]. Therefore, maintaining an efficient mitochondrial function for ATP production and bioenergetics is required for efficient MC function and retinal homeostasis. An injury to the neurons and/or microvasculature leads to MC activation/reactivity or gliosis, to prevent further neurovascular injury as manifested by the expression of the radial glial fibrillary acidic protein (GFAP) [7,8]. However, prolonged MC activation leads to aberrant gene expression and retinal injury by producing inflammatory cytokines, chemokines, and growth factors, such as NLRP3 inflammasome, IL-1β, TNF-1α, VEGF, and others [9].

Thioredoxin-interacting protein (TXNIP) is one of the genes which is strongly induced early in diabetes and by high glucose in all tissues examined so far, including retinal and renal cells [10,11,12]. TXNIP is defined as a pro-oxidative stress, pro-inflammatory, and pro-apoptotic protein induced in diabetes and its complications. TXNIP’s action includes binding to and inhibition of the anti-oxidant and thiol, reducing the capacity of thioredoxin (Trx), causing cellular oxidative/nitrosative stress (ROS), ER-stress, and sterile inflammation [13,14,15,16,17]. Trx1 and its reductase TrxR1 are expressed in the cytosol and nucleus, while Trx2/TrxR2 are the mitochondrial isoforms. TXNIP is present in all cellular compartments, including the cytoplasm, nucleus, and mitochondria, thereby, influencing various organelles functions. We have shown that TXNIP is highly induced in the diabetic rat retina, and that TXNIP knockdown by siRNA prevents early molecular abnormalities of DR, which include inflammatory gene expression, capillary basement membrane thickening, neuronal injury, radial GFAP expression, as well as LC3B punctate accumulation [12]. In in vitro cultures, high glucose-induced TXNIP expression causes retinal endothelial cell inflammation and chromatin remodeling [13], pericyte oxidative stress and apoptosis [18], MC inflammation [19], and RPE dysfunction [20]. TXNIP expression remains elevated as long as hyperglycemia persists, while insulin and insulin-like growth factor 1 (IGF-1) reduce TXNIP expression, however, it lasts for a few hours only if hyperglycemia persists [11]. Therefore, we propose that TXNIP is an excellent target for gene and drug therapy in DR.

Damaged or dysfunctional mitochondria produce less ATP while generating ROS, mediating mitochondrial oxidative stress, bioenergetics failure, and cell death [14,20]. Under sustained hyperglycemia, TXNIP migrates to the mitochondrion and inhibits the Trx2/TrxR2 anti-oxidant system [21,22]. We have shown that TXNIP is involved in mitophagic flux and lysosome enlargement in RPE cells under high glucose environment [20]. However, the mitophagic flux and lysosomal dysfunction in retinal MC are still unexplored. Mitophagy is a complex process involving separation of damaged parts of the mitochondrion by fission using dynamin-related protein 1 (Drp1) and fission protein 1 (fis1), followed by ubiquitin tagging by the Pink1–Parkin pathway [23,24]. Pink1 (PTEN-induced kinase 1) is a mitochondrial inner membrane kinase, which is degraded under the functioning mitochondria [25]. However, upon mitochondrial membrane damage/depolarization, Pink1 is retained in the outer mitochondrial membrane and phosphorylates outer membrane proteins, such as VDAC1 (voltage-dependent anion-selective channel 1) and Mfn2 (mitofusin 2), which are recognized by Parkin, an E3 ubiquitin ligase, for ubiquitination [16,26,27]. Then, mitophagy adaptors optineurin/p62-sequestosome1 and LC3B-II (microtubule-associated proteins 1A/1B light chain 3B) in double membrane autophagophore form the mitophagosome, inside of which damaged mitochondria are enclosed [20,28]. Subsequently, mitophagosomes fuse with lysosomes for degradation and exocytosis. Mitophagy, in fact, is a survival process for maintaining cellular homeostasis, bioenergetics, and viability. However, excess mitophagy causes a reduction in mitochondrial number and bioenergetics failure, while lysosome enlargement and lysosomal membrane permeabilization (LMP) lead to leakage of digestive enzymes to the cytosol [20,29]. Conversely, a slow mitophagy leads to the accumulation of damaged mitochondria releasing mtROS and oxidized mtDNA, which activate NOD-like LRR and pyrin domain-containing protein 3 (NLRP3) inflammasome and caspase-1 dependent interleukin IL-1β/IL-18 processing and pyroptosis via gasdermin-D mediated plasma membrane pore formation [14,30]. Thus, maintaining an optimal mitophagic flux, mitogenesis, and lysosome biosynthesis is needed. However, both mitophagic flux and mitogenesis are dysregulated in various chronic neurodegenerative diseases, including retinal diseases of DR, glaucoma, and age-related macular degeneration (AMD).

Mitophagy is a dynamic process and its optimal flux in physiology and pathophysiology is unknown at present. Hence, developing molecular probes for studying mitophagy in cells in in vitro cultures and in in vivo animal models of diseases is critically important. The aim of this study is also to develop a new mitophagic probe that can be used both in in vitro and in vivo systems. In fact, mitophagic probes have been developed before, including mitochondrial matrix-targeted mt-Keima, a coral protein that emits green light inside mitochondrial at neutral or alkaline pH (>7.0), while it emits red light in acid lysosomes (pH 4.5–5) after the mitophagic flux [27,31,32,33]. However, mt-Keima is detected in live cells alone as lysosomal pH loses upon fixation, and, therefore, mt-Keima emits green both in mitochondria and lysosomes. Hence, its use in animal models is limited as measuring fresh (live) tissues with mt-Keima may be inconvenient or not possible for handling multiple samples at a time for delicate tissues, such as the retina. On the other hand, another mitophagy probe has been developed, known as mito-QC (mitochondrial quality control), using co-expression of mCherry (red) and GFP (green) linked with an outer membrane protein fis1 N-terminus sequence, which inserts into the outer mitochondrial membrane [32]. In mitochondria, this mitophagic probe appears yellow (red and green co-localization), while in acidic lysosomes GFP fluorescence is quenched, therefore, appearing red after the mitophagic flux. One disadvantage of the fis1-mCherry-GFP mito-QC may be that the outer mitochondrial membrane probe is also degraded by proteases/proteasomes, which subdue the signal [27]. In addition, the insertion of fis1-mCherry-GFP in the outer mitochondrial membrane may interfere with mitochondrial dynamics of fission and fusion processes. For these reasons, we developed an adenovirus-CMV-2×mt8a-mCherry-EGFP (2×mt8a-CG) using two mitochondria target sequences (MTS) in tandem to express inside the mitochondrial matrix. Earlier, we have utilized the Ad-CMV-mt-Keima probe in our studies [20]. Using these two mitophagic probes, we examined the mitophagic flux in a retinal Müller cell line (rMC1) under high glucose conditions and the effect of treatment with triple drug combination, targeting TXNIP and the mitochondrial–lysosomal axis. Our results showed that high glucose induces TXNIP expression, redox stress, and mitophagic flux in rMC1, which are prevented by treatment with the triple drug combination, which may have potential in DR therapy.

## 2. Materials and Methods

### 2.1. Cell Culture

rMC1 cells were propagated in medium containing LG (1 g/L) DMEM/F12 (4:1 ratio) ((DMEM cat#10-014-CM, purchased from Mediatech Inc. (Manassas, VA, USA) while F12 was from HyClone (Logan, UT, USA, cat# SH30026.01)), 5% fetal bovine serum (cat# MT35010CV, from Corning (Corning, NY, USA)), 100 U/mL of penicillin, and 100 μg/mL of streptomycin at final concentration (from Hyclone). After reaching ~60% confluence, the cells were maintained in 1% serum with either low glucose (LG, 5.5 mM) or high glucose (HG, 25 mM) for 3 days. MitoSOX Red (M36008), MitoTracker Red CMXROS (M7512), and JC1 dye (cat #T3168) were obtained from Life Technologies (Carlsbad, CA, USA) and used to detect mitochondrial stress. Drugs used in this study were purchased from commercial vendors, such as TXNIP-IN1 (Cat# HY-115688, MedChemEXpress, Monmouth Junction, NJ, USA) and Mito-Tempo (Cat# SML0737), and ML-SA1 (Cat# SML0627) and Tranilast (Cat# T0138) from Sigma, St. Louis, MO, USA. Amlexanox (A6733), while SS-31 (Cat#1757721) was from Anaspec (Fremont, CA, USA). TXNIP knockout by CRISPR/Cas9 and gRNA (T1 + 3) has been described before [21], and they were maintained in HG or LG for 3 days in DMEM/F12 medium, as described above.

MitoSOX and JC1 assay: these assays were performed similar to those described before [21]. Briefly, the mitochondrial ROS formation in rMC1 was performed with MitoSox dye, which permeates mitochondria and produces red fluorescence upon oxidation by superoxides. Approximately 5 × 10^3^ cells/mL were seeded in 48-well plates and exposed to HG or LG for 3 days. Then, the cells were washed with PBS and afterwards, 5 μM MitoSox was incubated for 10 min at 37 °C. The cells were further washed with PBS (3 times) and fluorescence was measured at EX510/Em590 using a Gemini Fluorescence Microplate reader (Molecular Device, Sunnyvale, CA, USA). For the mitochondrial membrane potential measurement, JC1 dye was added, similar to that described above for MitoSox. JC1 enters the cell and accumulates in mitochondria as orange-red aggregates, with EX535/Em590. Upon mitochondrial oxidative stress and membrane depolarization under HG, monomer JC1 leaks out of mitochondria and emits green fluorescence. The decrease in red fluorescence is measured as mitochondrial membrane depolarization in rMC1 under HG versus LG.

### 2.2. Live-Cell Imaging

For live-cell imaging of mt-Keima and mCherry-EGFP, rMC1 cells were cultured in six-well plates in coverslips. They were maintained under LG or HG conditions for 3 days (21). In addition, 2 µL of Ad-CMV-mt-Keima or Ad-CMV-2×mt8a-mCherry-EGFP (named 2×mt8a-CG), with an ~4.8 × 10^10^ PFU/mL GC titer (VectorBuilder, Malvern, MA, USA) was transduced for 3 days before capturing the images. Inhibitors were added 24 h before imaging. The medium was changed to HBSS solution without phenol. A Zeiss LMS 780 Confocal Microscope (Jene, Germany) fitted with a CO_2_ chamber and live-cell imaging platform was used to capture images in triplicate cultures, as previously described [20]. (i) For the green emission of mt-Keima in mitochondria, we used a 458 nm argon laser at 5% laser power and our emission collection window was set from 570–694 nm. For the red mt-Keima in lysosomes, we used a 561 nm HeNe laser at 5% laser power and the emission collection window was similar to that of the green signal (570–694 nm). (ii) For the green EGFP in 2×mt8a-CG, we used a 488 nm argon laser at 6% power and our emission collection window was from 499–552 nm. For mCherry (red) in 2×mt8a-CG, we used a 561 nm HeNe laser at 5% power and our emission collection window was from 575–659 nm. A Zeiss LMS 780 Confocal Microscope was used to capture multiple images in triplicate at 63X/1.4NA objective. In each experiment, all images were captured within 1–1.30 h. Images were analyzed using ZEN 3.0 blue lite software and compiled in Adobe Photoshop.

### 2.3. Fixed-Cell Imaging

For imaging fixed cells, rMC1 cells were cultured with 2 mL of DMEM:F12 medium in six-well plates, containing three sterile coverslips in each well. Similar to live-cell imaging conditions, LG and HG conditions were maintained for 3 days, while 2 µL of the Ad-CMV-2×mt8a-CG with an ~4.8 × 10^10^ PFU/mL GC titer were added. Media changed after 2 days, and cells were fixed in 4% PFA (paraformaldehyde) in PBS, pH 7.0, and mounted with DAPI to stain nuclei on day 3. In addition, we also used TXNIP (cat#14714S, Cell Signaling), TrxR2 (sc-166259, Santa Cruz and 12029S, Cell Signaling), TFEB (32361S, Cell Signaling), and PGC1α (s518025, Santa Cruz) antibodies to stain fixed rMC1 cell experiments. Corresponding fluorescence-labeled secondary antibodies (red, Alexa 594 and green, Alexa 488) were used. A Zeiss LMS 780 Confocal Microscope was used to capture multiple images in triplicate at 63X/1.4NA objective. For the blue signal (DAPI in cell nuclei), we used excitation at 405 nm diode laser at 10% laser power and emission from 415–470 nm. For the EGFP, we used excitation at 488 nm argon laser at 6% power and our emission collection window was from 499–552 nm. For the red (mCherry), we used a 561 nm HeNe laser at 5% power and our emission collection window was from 575–659 nm. Images were analyzed similar to those described for live-cell imaging above. For immunofluorescence, prolong gold antifade reagent with DAPI (mounting medium, cat # P36935) was obtained from Molecular Probes (Eugene, OR, USA). Slides and coverslips were purchased from Fisher Scientific (Waltham, MA, USA).

### 2.4. Mitochondrial Isolation

Cellular fractionation was performed as previously described [21]. Briefly, rMC1 cells from 15 cm culture plates were scraped and washed with PBS and resuspended in mitochondrial isolation buffer (3 mM HEPES–KOH (pH 7.4), 210 mM mannitol, 70 mM sucrose, 0.2 mM EGTA, and protease inhibitor cocktail). Cells were then homogenized using a dounce homogenizer (~90 strokes). Several centrifugation steps were performed. First, cells were centrifuged at 2000 rpm for 5 min to separate the nuclear fraction and the supernatant was further subjected to 5000 rpm for 5 min to clear any debris. The clear supernatant was collected and centrifuged at 13,000 rpm for 15 min at 4 °C to obtain mitochondria in pellet. (i) The upper supernatant was collected as the cytosolic fraction. (ii) The pellet was washed twice with PBS and then re-suspended in RIPA buffer (Cat# R0278, Sigma) with protease and phosphatase inhibitors (Cat# MSSAFE, Sigma), sonicated briefly and spun at 2000 rpm for 5 min. Finally, the supernatant was used as the mitochondrial fraction. Protein concentrations were estimated by Bradford assay.

### 2.5. Western Blotting

Western blotting (WB) was performed as previously described [20,21]. Briefly, 30 μg of protein from cytosolic and mitochondrial fractions was loaded for SDS-PAGE, and WB analysis was performed. Primary antibodies TXNIP (Cat#14714S), Tom20 (Cat# 42406S), TFEB (32361S) from Cell Signaling Technology, and PGC1α (s518025, Santa Cruz) and LC3B (L10382, Pierce) were employed. Actin-β (Sc-1616, Santa Cruz) or LDHA (Cat# PA5-27406, Invitrogen) was used to normalize protein bands. Primary antibodies were incubated at 1:1000 in 5% non-fat milk in Tris-buffer saline, pH 7.4 at 4 °C overnight in a rotor, while HRP-labeled secondary antibodies were incubated at 1:2000 or 1:3000 dilution in 5% non-fat dry milk for 2 h at room temperature with rotation. The Clarity Western ECL Substrate (Cat#1705061, Biorad) detected the immunoreactive proteins.

### 2.6. Diabetes Induction in Rats and Immunohistochemistry

Diabetes induction and intravitreal siRNA injection in adult male Sprague–Dawley rats (∼275 g) were performed as described previously [8,20]. The rats were treated in accordance with the principles outlined in the NIH Guidelines for the Care and Use of Laboratory Animals, and the protocol was approved by the Institutional Animal Care and Use Committee (WSU IACUC). At the end of diabetes duration (2 months), the eyes were removed and processed for immunohistochemistry. Retinal sections were blocked for 1 h in a PB solution that contained 5% Chemiblocker (Chemicon, Temecula, CA, USA), 0.5% Triton X-100, and 0.05% sodium azide. SNO antibodies (N54111) were from Sigma. GFAP monoclonal antibody labeled with Alexa Flour 488 (Cat# 53-9892-82) was from ThermoFisher Scientific. The primary antibodies were diluted in the same solution (1: 200 dilution) and applied overnight followed by appropriate secondary antibodies (1:600 dilutions) conjugated with Alexa Fluor 488 or 594 for 2 h for s-nitrosylated proteins. DAPI was used to stain nuclei. The images were captured at 40× objective lens and eyepiece magnification 10× (overall magnification of 400×) in an OLYMPUS BX 51 Fluorescence microscope (Center Valley, PA, USA), which was fitted with a triple DAPI/FITC/TRITC cube, a DP70 digital camera, and image acquisition software.

### 2.7. Statistical Analysis

The results are expressed as the mean ± SEM of the indicated number of experiments. Comparisons between two sets of experiments were analyzed using the unpaired two-tailed t-test, whereas one-way ANOVA followed by the Bonferroni post hoc test was used to determine differences among means in multiple sets of experiments. A *p*-value of <0.05 was considered to be statistically significant.

## 3. Results

### 3.1. Diabetes Induces TXNIP Expression, Redox Stress, and Müller Cell Activation in the Diabetic Rat Retina

Diabetes induces TXNIP expression and redox stress in the retina, as demonstrated by enhanced staining of TXNIP and protein s-nitrosylation (SNO) when compared to non-diabetic control retinas (Figure 2A,B), indicating that TXNIP upregulation and an inhibition of the Trx/TrxR redox system may be involved in this process. In addition, enhanced expression of extracellular matrix protein Tenascin-C was observed at the blood vessel basement membrane and GCL/ILM (Figure 2C). Tenascin-C is associated with tissue injury and repair, and involved in angiogenesis and innate immune responses (via TLR4 binding) under cell stress. Furthermore, autophagosomes, LC3B punctate formation, are seen in the diabetic rat retina more than in the normal rat retina (Figure 2D). MCs react to retinal injuries in an effort to prevent further retinal damage, which is known as Müller cell reactivity or gliosis, and manifested in diabetic retinas by an induction of radial GFAP in the neuroretina (Figure 2E).

However, prolonged Müller cell activation or injury itself (such as a decrease in glutamine synthetase, GS, Appendix A) leads to release of inflammatory cytokines, chemokines, and excitatory metabolites, which may be harmful to retinal neurons and microvasculature. GS was observed to have less staining in the soma of MC in the INL (Appendix A, red arrows), while it appears to be concentrated at the ILM of MC end feet or at the neurofilament layer of ganglion cells (GCL, Appendix A, yellow arrows). Such an aberrant redistribution of GS in the ILM may suggest neuronal/axonal injury in DR because GS is required for the conversion of neurotransmitter glutamate to glutamine. We have previously shown that TXNIP knockdown by siRNA in diabetic rat retinas reduces several early retinal abnormalities, including capillary basement membrane thickening, neuronal injury, and GFAP expression [8]. We also observed here that TXNIP silencing by siRNA reduces TXNIP expression, LC3B punctate formation, protein-SNO (Appendix A), and GFAP (Appendix A) in the diabetic rat retina.

### 3.2. High Glucose Induces TXNIP Expression, Mitochondrial Stress, and Mitophagic Flux in rMC1 in Culture

Müller glial cells are metabolically active cells, and maintain their bioenergetics through both cytosolic glycolysis and mitochondrial oxidative phosphorylation. However, under sustained hyperglycemia and DR, TXNIP expression, redox stress, and mitochondrial dysfunction may occur. We showed in this study that TXNIP expression, protein-SNO (Figure 3A,B), mitochondrial oxidative stress, and membrane depolarization (Figure 3C–E) are promoted by HG treatment for 3 days in rMC1. Furthermore, mitochondrial fragmentation and mitophagic flux were increased in rMC1 by HG than in LG, which was seen as yellow mitochondria and red mitolysosomes in Ad-CMV-2×mt8a-CG transduced rMC1 (Figure 4A). (The expression vector construct for Ad-CMV-2×mt8a-mCherry-EGFP is as shown in Appendix A). As mentioned above, in this construct, intact mitochondria show yellow, a combination of red mCherry, and green EGFP, while lysosomes are seen as red (mCherry) after mitophagic flux due to EGFP quenching in acidic environment (pH 4.5–5). Previously, we showed that TXNIP knockout by CRISPR/Cas9 and TXNIP gRNA prevents high glucose-induced mitochondrial redox stress and mitophagic flux in retinal MC (21). This observation is further supported in this study by using Ad-CMV-2×mt8a-CG vector in TXNIP knockout rMC1 (T1 + 3), which is shown in Figure 4B. Similarly, mitophagy induction by CCCP (20 μM, 24 h) in rMC1 was reduced by TXNIP knockout in rMC1 (T1 + 3) using the mt-Keima probe (Appendix A). In addition, mitochondria-targeted anti-oxidant SS-31 and lysosomal agonist ML-SA1 also reduced mitophagic flux.

### 3.3. Combination Drug Treatment Reduces TXNIP Level and Mitophagic Flux in rMC1 under High Glucose Conditions

We hypothesized that under chronic metabolic stress and disease, the cellular and molecular abnormalities may continue once the disease process has begun, even if the initial insult has diminished. Therefore, instead of targeting TXNIP alone, combination drugs targeting TXNIP and the mitochondrial–lysosomal dysfunction under a high glucose environment will maintain cellular homeostasis. For this approach, we employed a recently developed inhibitor of TXNIP-Trx interaction (TXNIP-IN1), Mito-Tempo (mitochondria-targeted anti-oxidant) and ML-SA1 (lysosome-targeted MCOLN1 and TFEB activator), the rational for which is shown in Appendix A.

We treated rMC1 cells individually or in combination of the three drugs under LG and HG conditions. TXNIP-IN1 itself did not have an effect on protein levels of TXNIP, Trx1, or Trx2 (Appendix A); however, TXNIP levels were altered by drugs that modify histone acetylation (Appendix A). We further observed, on Western blots, that Mito-Tempo (5 μM) reduces TXNIP protein level under HG, while TXNIP-IN1 (5 μM) or ML-SA1 (20 μM) does not have an effect on TXNIP (Figure 5A). Conversely, a combination of the three drugs (designated as TMM) reduces TXNIP level in both cytosolic and mitochondrial fractions (Figure 5A,B). Similarly, in immunofluorescence staining of rMC1, HG increases TXNIP staining, while reducing redox protein TrxR2 more than that observed in LG (Figure 6, upper two panels). Conversely, when rMC1 cells are treated with TMM, the level of TXNIP is reduced in HG, while TrxR2 is enhanced in both LG and HG conditions (Figure 6, lower two panels). Therefore, we used the TMM combination in further experiments to examine their effects on mitophagic flux to lysosomes.

For these experiments, we used Ad-CMV-2×mt8a-CG both in live and fixed-cell confocal imaging. In live-cell imaging under LG, as mentioned before, mitochondria are seen as yellow, a combination of mCherry and EGFP, while lysosomes are seen as red spots, mCherry alone (Figure 7, first panel). However, under HG, some of the yellow mitochondria are seen as fragmented and enlarged red mitolysosomes, indicating a mitophagic flux (Figure 7, second panel). Furthermore, we used another three drugs in combination that we have previously shown effective in reducing auronofin-mediated redox stress and mitophagic flux in retinal pigment epithelial cells [20], shown in Appendix A. These drugs include amlexanox (4 μM), which is an TBK1 kinase inhibitor that phosphorylates optineurin and p62 adaptors to increase their activity, SS-31 (4 μM, a mitochondria-targeted anitoxidant), and Tranilast (56.6 μM, an inhibitor of NLRP3), denoted as AST, and they together also inhibit TXNIP expression (Appendix A, inset). We compared this triple drug combination of AST with the current triple combination of TMM. We observed that both AST and TMM reduce mitophagic flux in rMC1 in HG, as revealed by fewer and reduced sizes of the red mCherry lysosomes (Figure 7, lower two panels), which is more or less similar to that observed in LG. However, TMM appears to maintain more filamentous mitochondria than AST treatment. A similar observation is also observed with fixed-cell confocal imaging of Ad-CMV-2×mt8a-CG transduced rMC1 (Figure 8), indicating that 2×mt8a-CG probe can be used both in live- and fixed-cell imaging.

### 3.4. Combination TXNIP-IN1, Mito-Tempo, and ML-SA1 (TMM) Treatment Induces Transcription Factor TFEB and PGC1α Nuclear Localization in rMC1 Cells under High Glucose Conditions

Lastly, we examined if the triple drug combination (TMM) targeting TXNIP and the mitochondrial–lysosomal axis has an effect on transcription factor nuclear localization of TFEB (master regulator of lysosomal biogenesis and autophagy) and PGC1α (peroxisome proliferator-activated receptor-gamma coactivator 1alpha, mitogenesis factor) in rMC1 under HG and LG conditions. Here, we briefly provide a rationale for looking into these two factors. TFEB is needed for maintaining lysosomal function, biogenesis, and autophagy gene expression while PGC1α is involved in nuclear gene expression for mitochondrial biogenesis [34,35]. TFEB is trapped in the cytosol by phosphorylation (at Ser211) with mTORC1 via binding to scaffolding protein 14-3-3 [36,37].

Dephosphorylation by calcineurin phosphatase releases TFEB from 14-3-3 and translocates to the nucleus for transcriptional activation of the CLEAR (coordinated lysosomal expression and regulation) gene network, which is involved in lysosome biogenesis, exocytosis, and autophagic gene expression (ATGs) to maintain a critical level of lysosomes and autophagic activity. In addition, TFEB also activates PGC1α and Nrf2 [36], which are transcription actors needed for mitogenesis and anti-oxidant gene expression. Therefore, the activation of TFEB and PGC1α may be important for maintaining the mitochondrial–lysosomal axis and an efficient anti-oxidant system in cells. In the present study, as shown by the results in Figure 9 in the upper two panels and in Appendix A*,* HG reduces nuclear localization of both TFEB and PGC1α in rMC1 more than in LG, while treatment with TMM increases nuclear localization of both factors in LG and HG (Figure 9, lower two panels), suggesting maintenance of the mitochondrial and lysosomal function by treatment of these drugs together. Further studies are required to investigate the beneficial effects of this combination treatment in cellular and animal models of DR and age-related retinal diseases.

## 4. Discussion

Damaged mitochondria produce less ATP while continuing to generate ROS. Therefore, removal of damaged mitochondria by mitophagy is critical for cell homeostasis and survival. In addition, replenishment of new mitochondria (mitogenesis) and biosynthesis of lysosomes and autophagic genes are essential to maintain an effective autophagy/mitophagy and bioenergetics. However, both mitophagy/mitogenesis and lysosomal function are deregulated in chronic diseases, including DR and various age-related retinal and neuronal diseases. Therefore, maintaining the mitochondrial–lysosomal axis may be critically important to prevent cell distress, dysfunction, and disease progression. In this study, we provide evidence that (i) retinal injury and Müller cell activation occur in early DR, and TXNIP knockdown by siRNA prevents molecular alterations in DR. In addition, Müller cell redox stress and mitochondrial dysfunction are seen under HG in in vitro cultures. (ii) A triple drug combination treatment, comprising of TXNIP-IN1, Mito-Tempo, and ML-SA1 (TMM), targeting TXNIP and mitochondrial–lysosomal stress are effective in normalizing mitophagic flux and transcription factor TFEB and PGC-1α nuclear translocation in rMC1 under a high glucose environment, as seen in diabetes. A summary of the potential mechanism(s) for the TXNIP and redox stress-induced organelle injury, Müller cell activation, and pathogenesis of DR, and potential interventional opportunities are depicted in Figure 10 as a summary. (iii) A newly developed mitophagy probe, Ad-CMV-2×mt8a-CG, will be an important tool to study mitophagic flux in both live and fixed cells in in vitro cultures, as well as in in vivo animal models. Previously, we showed that mt-Keima is an excellent mitophagic probe to measure mitophagic flux to lysosomes in live cells [14,20,23]; however, its use in fixed cell and tissue is limited.

TXNIP and its associated oxidative/nitrosative stress, mitochondrial dysfunction, mitophagy deregulation, and lysosome destabilization are seen in retinal cells under high glucose conditions [20,21]. Therefore, TXNIP and redox stress may play critical roles in initiating disease process and progression in DR. Hence, we proposed that TXNIP is an important target for reducing cellular oxidative stress, inflammation, and premature cell death [8,11]. Nonetheless, in chronic diseases, such as DR and age-related retinal diseases, including AMD, once the disease process has begun, then downstream organelle dysfunction of mitochondria and lysosome (and other organelles, such as endoplasmic reticulum and peroxisomes) may continue in a feed-back or feed-forward manner, independently of the initial insult. In addition to TXNIP and the Trx/TrxR redox system, other ROS generating systems (NADPH oxidase and Xanthene oxidase) and anti-oxidants (GSH, SOD1, and SOD2) may also be involved in dysregulation of the mitochondrial–lysosomal axis in diabetes and under glucolipotoxicity [38]. Therefore, we propose a triple drug combination treatment targeting TXNIP and downstream mitochondria–lysosome pathway to neutralize/normalize cellular redox stress and mitophagic flux. Of these, we show that TMM treatment restores mitophagy and nuclear translocation of transcription factors TFEB and PGC1α under a high glucose environment (Figure 7, Figure 8 and Figure 9), which is important for maintaining the mitochondria–lysosome axis and autophagy process. During autophagy/mitophagy, not only aggregated proteins or damaged mitochondria but also ATGs and adaptor proteins accompany autophagosomes to lysosomes for degradation. Therefore, their replenishment is also important for maintaining an active autophagy/mitophagy process. Transcription factor TFEB is a master regulator of lysosome biogenesis, lysosomal enzyme expression, and ATG gene expression [39]. TFEB also induces the expression of transcription factors, PGC1α and Nrf2, which are critical for nuclear-encoded mitochondrial and anti-oxidant genes. PGC1α itself also induces TFEB expression in a coordinate manner. PGC1α gene targets include mitochondrial transcription factor TFAM and Nrf1, important for mitochondrial gene expression [40,41,42].

The question of mitophagy being beneficial or detrimental in chronic diseases is still unknown [23,43]. Mitophagy itself is a cellular protective mechanism by removing damaged mitochondria, particularly in acute injury [44]. However, in chronic diseases, a continued redox stress, mitochondrial dysfunction, and excess mitophagic flux will result in reduction in mitochondrial number and bioenergetics deficiency, while lysosomal overloading causes its enlargement and LMP [20]. Conversely, a slow mitophagy with dysfunctional lysosome will cause an accumulation of damaged mitochondria, redox stress, loss of cytosolic sanctity, inflammation, and premature cell death by apoptosis or other cell death mechanisms, including pyroptosis and ferroptosis [14,45]. Damaged mitochondria release mtROS and oxidized mtDNA, which are recognized by cytosolic pattern recognition innate immune receptors, such as NLRP3 inflammasome, thereby, activating caspase-1, IL-1β/IL-18, gasdermin-D/E, and pyroptosis [14]. Furthermore, the release of lysosomal proteases, including Cathepsin B, L, or D, lead to processing of pro-caspase-1 directly [14]. Cathepsin B may also act on mitochondrial outer membrane proteins and cause mitochondrial dysfunction [46].

In regards to the importance of the removal of damaged mitochondria, mitophagy itself is a process to block NLRP3 inflammasome activation and sterile inflammation. Therefore, the role of Pink1/Parkin in the mitochondrial quality control has been investigated intensively in neurodegenerative diseases, including Parkinson’s and Alzheimer’s [47,48]. Parkin is an E3 ubiquitin ligase involved in the regulation of mitophagy, mitogenesis, and NLRP3 inflammasome resolution [47,49]. (i) Parkin ubiquitinates mitochondrial outer membrane proteins, such as VDAC1 and Mfn2, in damaged mitochondria and mark for autophagophore recognition by adaptors optineurin/p62 and LC3BII. (ii) Parkin also ubiquitinates PARIS (PARkin-interacting substrate) for proteasomal degradation [50]. PARIS binds to and entraps PGC1α in the cytosol, thereby preventing its migration to the nucleus. Parkin-mediated degradation of PARIS releases PGC1α and enhances nuclear translocation and gene transcription. (iii) We have previously shown that TXNIP is involved in transcriptional activation of NLRP3 and pro-IL-1β via enhanced NF-κB nuclear localization [11,12,18]. Furthermore, under excess ROS stress, TXNIP dissociates from Trx1 and assembles the macromolecular NLRP3 inflammasome comprising of NLRP3, ASC, and pro-caspase-1. However, the feedback regulation of the NLRP3 inflammasome, its degradation, and inflammation resolution are still to be revealed. In this regard, Parkin has been suggested to be involved in NLRP3 inflammasome ubiquitination and its autophagic degradation involving p62 and LC3BII autophagosome [51,52]. We also observed that Parkin knockdown by siRNA reduces mitophagy, but mitochondrial fragmentation remains in rMC1 under HG while TFEB and PGC1α nuclear translocation is prevented (data not shown). We will investigate the role of Parkin and its partner PINK1 in mitophagy and mitochondrial quality control in a separate study.

In conclusion, our study shows that TXNIP expression, redox stress, and mitophagy dysregulation may play an important role in retinal Müller cell dysfunction in DR. Gliotic MCs produce various inflammatory cytokines, chemokines, and altered growth factors, which are harmful to retinal neurons, microvascular, and RPE [53,54]. In fact, mitophagy is also enhanced in human Müller cell line MIO-M1 under high glucose in vitro and in Akita mice at 2 months of diabetes duration [55], probably as a protective effort [19]. Although mitogenesis machinery appears to remain unaffected, mitogenesis is defective in DR [40,55], which may be due to the inability of transcription factors to import into the nucleus (this study). Therefore, maintaining mitochondrial quality control via a reduction in oxidative stress, keeping an optimal level of mitophagic flux/mitogenesis and lysosome function, will be critical for retinal homeostasis and preserving the visual function in DR. In this regard, the triple drug combination (TMM) therapy targeting TXNIP and the mitochondrial–lysosomal axis may reduce retinal redox stress, mitophagic flux, and maintain bioenergetics in DR. Finally, the mitophagic probe, Ad-CMV-2×mt8a-CG, may serve as an excellent tool for investigating the physiological and pathological mitophagic flux in the retina and other tissues.

## Figures and Tables

**Figure 1 diseases-09-00091-f001:**
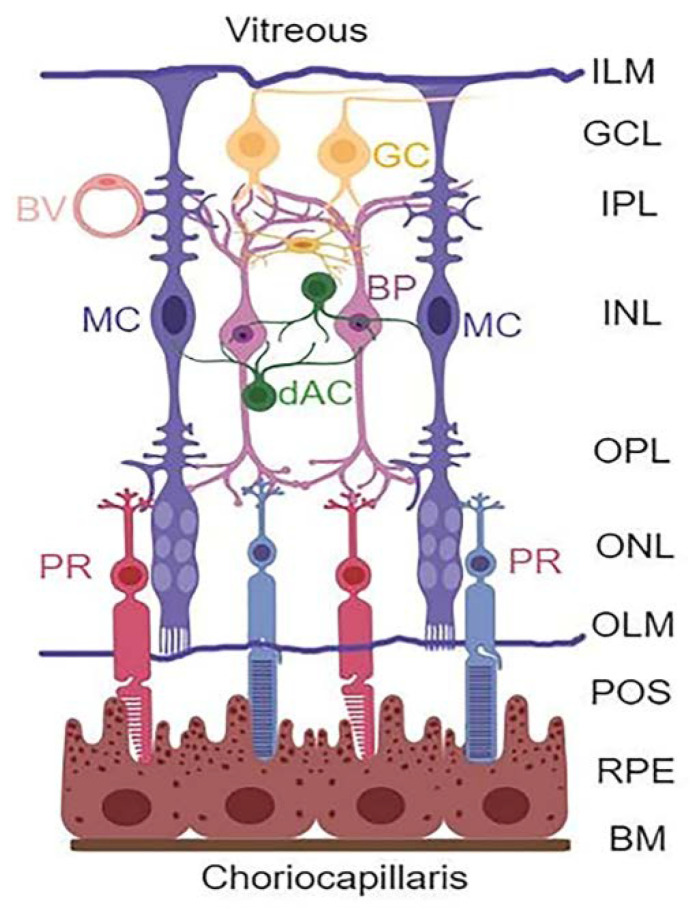
Retinal layers and cell types. The retina is structured in three main neuronal layers: ganglion cell layer (GCL), bipolar neuron layer (BP, INL), and photoreceptor layer (PR, ONL). Between these neuronal layers, axonal-synaptic connections are formed as the inner plexiform layer (IPL) between bipolar and ganglion cells, while the outer plexiform layer (OPL) is the connection between photoreceptors and bipolar neurons. Other retinal neuronal cells include amacrine dopamine (dAC) and horizonal cells. The retina also has macroglia (astrocytes and Müller glial cells) and microglia (immune cells). The astrocytes are found at the GCL or nerve fiber layer on the vitreous side, while MC occupy the entire length of the neuroretina forming inner limiting membrane (ILM) at the vitreous, and the outer limiting membrane (OLM) at the photoreceptor outer segment (POS). Müller cells play critical roles in the formation of neurovascular unit and retinal homeostasis. (Created with BioRender.com, accessed on 9 December 2021).

**Figure 2 diseases-09-00091-f002:**
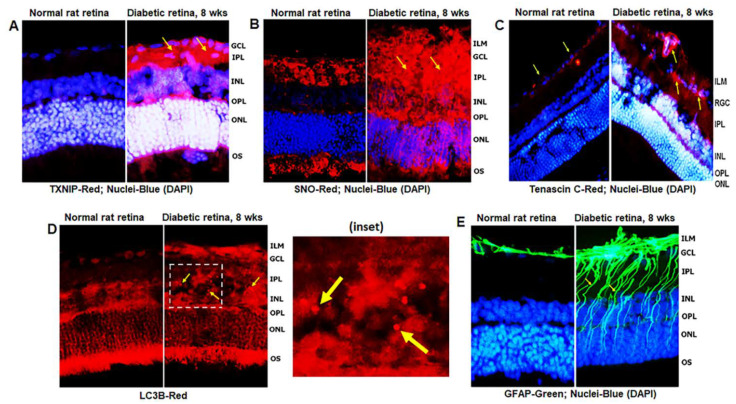
TXNIP expression and redox stress in the diabetic rat retina. (**A**) At 8 weeks of diabetes duration, TXNIP is strongly induced in the diabetic rat retina when compared to the non-diabetic normal rat retina. (**B**) TXNIP expression is associated with increases in protein-s-nitrosylation (SNO), an indication of retinal redox stress in diabetes, and (**C**) expression of extracellular matrix protein, tenascin C, in capillary basement membrane and GCL, suggesting vascular inflammatory response. (**D**) Diabetes also increases autophagosome punctate (arrows in inset) in the retina than in normal rats. (**E**) Müller cell activation (gliosis) is observed in the diabetic rat retina, as indicated by the expression of radial glial fibrillary acidic protein (GFAP), suggesting retinal injury and/or abnormalities in the diabetic retina. The images represent *n* = 3, taken by an Olympus BX51 fluorescence microscope at 400× magnification.

**Figure 3 diseases-09-00091-f003:**
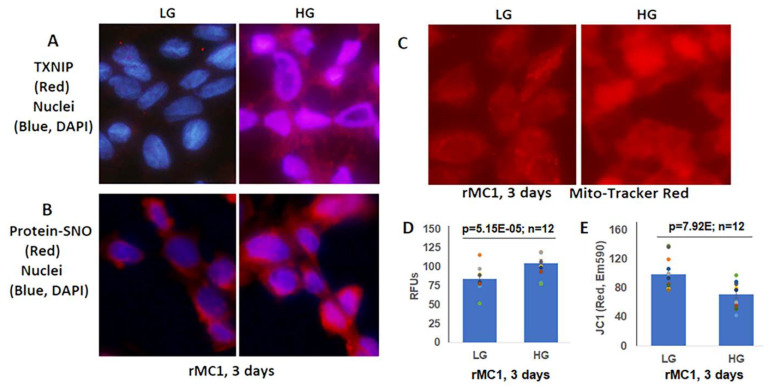
High glucose induces TXNIP expression and redox stress in rMC1 in in vitro culture. (**A**) HG exposure of rMC1 cells for 3 days increases TXNIP expression and (**B**) protein s-nitrosylation (SNO), suggesting cellular oxidative/nitrosative stress when compared to LG. DAPI was used to counter stain the nuclei (blue). (**C**,**D**) Similarly, mito-tracker red staining (EX579/Em599) and mitoSox fluorescence intensity (EX510/Em580) are also enhanced, indicating mitochondrial stress and damage. (**E**) In agreement, mitochondrial membrane potential is disturbed/depolarized (reduced red of JC-1 aggregates, EX535/Em590) in rMC1 under HG versus LG. The immunofluorescence images represent *n* = 3, captured by an Olympus BX51 fluorescence microscope at 400× magnification.

**Figure 4 diseases-09-00091-f004:**
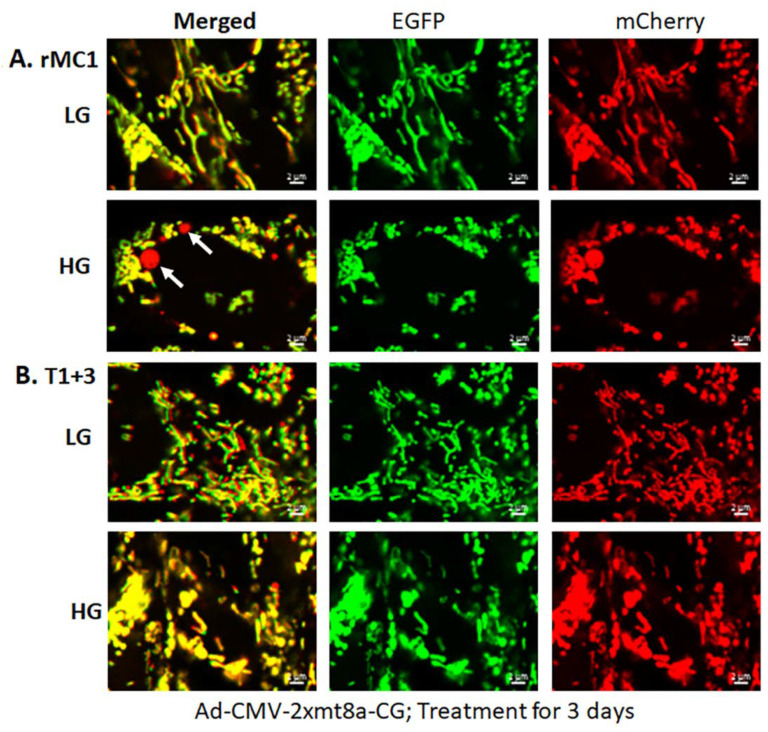
HG induces mitophagy in rMC1 and TXNIP knockout reduces mitophagic flux. (**A**) rMC1 cells were transduced with the mitophagy probe, 2×mt8a-CG (mCherry/EGFP), for 3 days in the presence of LG or HG. Live-cell confocal imaging shows that HG induces red mitolysosome formation (while arrows) when compared to LG. Active mitochondria emit red and green signals providing a yellow image, while EGFP signal loses in acidic lysosomes after the mitophagic flux. (**B**) TXNIP knockout by CRISPR/Cas9 and gRNA (T1 + 3) prevents mitophagic flux in rMC1 under HG. Images are representation of *n* = 3, captured by a Zeiss confocal microscope at 630× magnification.

**Figure 5 diseases-09-00091-f005:**
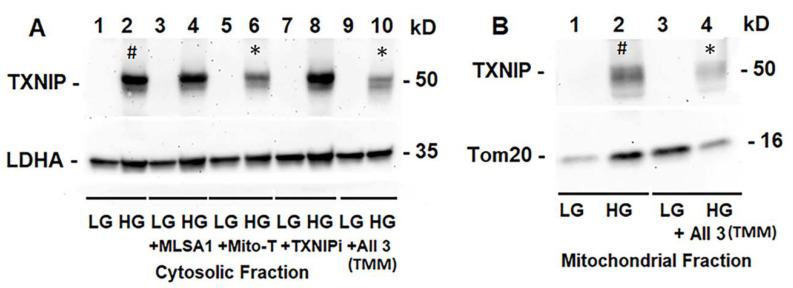
HG induces TXNIP expression and triple drug combination treatment reduces TXNIP in rMC1. (**A**) On Western blots, TXNIP expression increases ~4.62 +/− 0.34 folds by HG (lane 2 #, mean +/− SE versus LG, *n* = 3). Lysosome-targeted drug ML-SA1 (20 μM) or TXNIP-IN1 (TXNIPi, an inhibitor of TXNIP-Trx interaction, 5 μM) has no effect on HG-induced TXNIP expression, while mitochondria-targeted anti-oxidant mito-Tempo (5 μM) and triple drug combination, TMM, reduces TXNIP level by 69% (lane 6 *) and 50% (lane 10 *), respectively, versus HG (2), *n* = 3). (**B**) Similarly, TXNIP level also increases in the mitochondrion under HG (lane 2 #, ~2.3-folds versus LG, *n* = 3), while combination drug (TMM) treatment reduces ~70% (Lane 2 versus Lane 4 *, *n* = 3).

**Figure 6 diseases-09-00091-f006:**
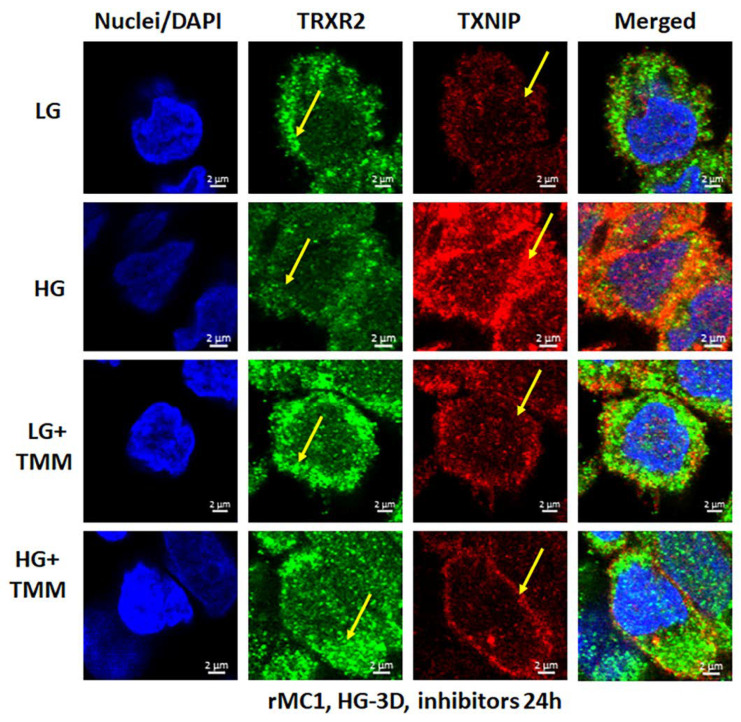
HG reduces mitochondrial thioredoxin reductase 2 (TRXR2) in rMC1. HG exposure of rMC1 for 3 days reduces TRXR2 level compared to LG, while that of TXNIP is increased (first two panels). On the other hand, treatment of rMC1 with triple drug combination TMM increases TRXR2 and decreases TXNIP level under both HG and LG, suggesting a beneficial effect of these drugs on HG-induced redox stress. A representative of *n* = 3 is shown from fixed-cell confocal imaging at 630× magnification in a Zeiss confocal microscope.

**Figure 7 diseases-09-00091-f007:**
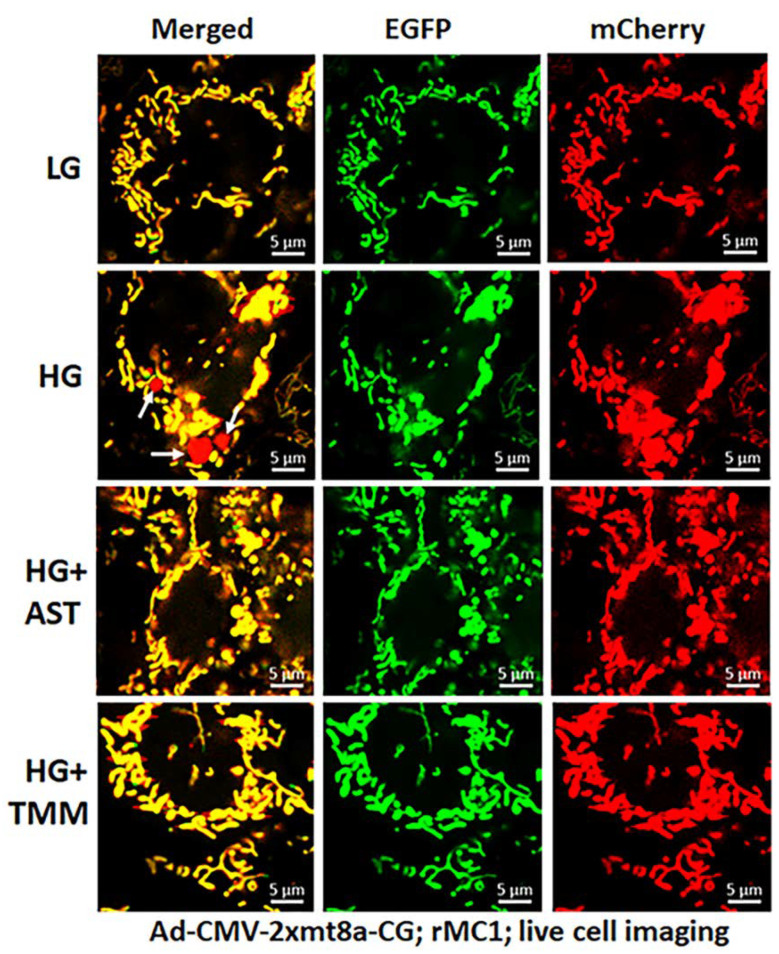
Combination triple drug treatment reduces mitophagy in rMC1 under HG in live-cell imaging. rMC1 cells were transduced with 2×mt8a-CG (mCherry-EGFP) for 3 days under HG or LG. When drug treatments were performed, they were present for 24 h before the confocal live-cell imaging. We observed that HG increases red mitolysosome formation (white arrows, HG second panel) when compared to LG. Active mitochondria show yellow (red and green), while damaged mitochondria after flux to lysosomes yield red due to EGFP (green) quenching. Interestingly, combination triple drug treatment with AST (amlexanox, SS-31 and tranilast) or with TMM (TXNIP-IN1+Mito-Tempo+ML-SA1) reduces mitophagic flux in rMC1 under HG (last two panels, respectively). A representative of *n* = 3 is shown here, captured at 630× magnification in a Zeiss confocal microscope.

**Figure 8 diseases-09-00091-f008:**
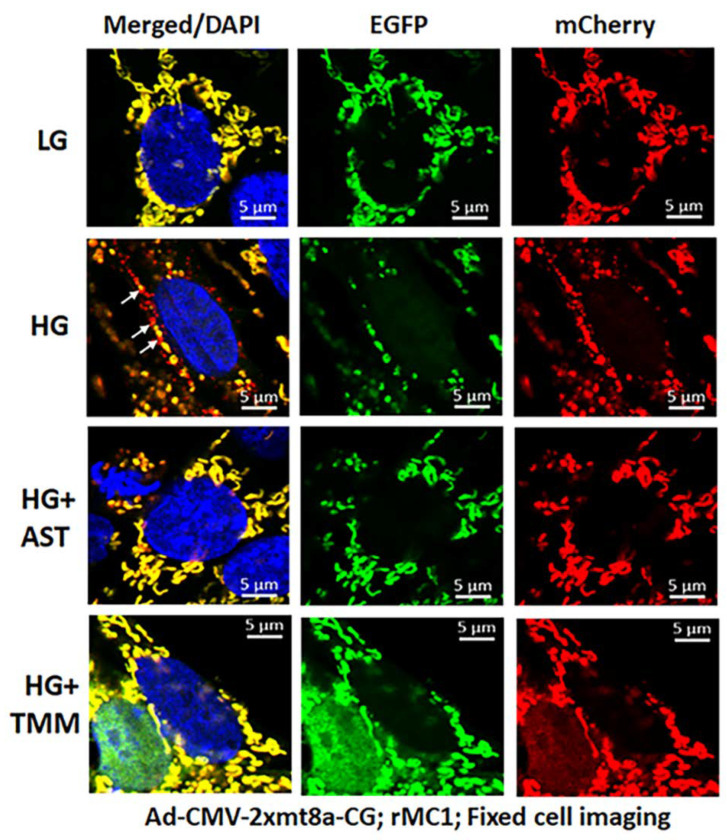
Combination triple drug treatment reduces mitophagy in rMC1 under HG in fixed-cell imaging. rMC1 cells were transduced with 2×mt8a-CG (mCherry-EGFP) for 3 days under HG or LG similar to live-cell imaging. When drug treatments were performed, they were added 24 h before fixing the cells. Nuclei were counter-stained with DAPI (blue). HG increases red mitolysosome formation (while arrows, HG second panel) in rMC1 cells than in LG. Active mitochondria show yellow (red and green), while damaged mitochondria emit red in lysosomes due to EGFP (green) quenching. AST (amlexanox, SS-31 and tranilast) or TMM (TXNIP-IN1+Mito-Tempo+ML-SA1) reduces mitophagic flux in rMC1 under HG (last two panels, respectively). A representative of *n* = 3 is shown here, captured at 630× magnification in a Zeiss confocal microscope.

**Figure 9 diseases-09-00091-f009:**
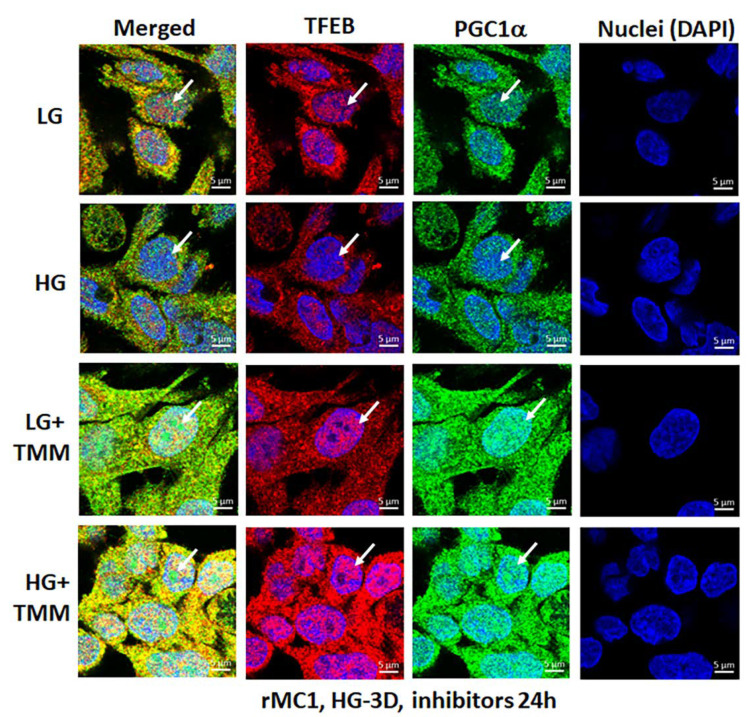
HG reduces transcription factor TFEB and PGC1α nuclear translocation in rMC1. Incubation of rMC1 with HG for 3 days reduces nuclear localization of both transcription factors, TFEB (red) and PGC1α (green) more than in LG (upper two panels). Nuclei is counter stained with DAPI (Blue). Interestingly, triple combination drug treatment (TMM) for 24 h, before ending the experiment, increases TFEB and PGC1α nuclear localization both in LG and HG, suggesting lysosomal and mitochondrial gene transcription and biogenesis. A representative of *n* = 3 in fixed-cell confocal imaging, captured at 630× magnification, in a Zeiss confocal microscope.

**Figure 10 diseases-09-00091-f010:**
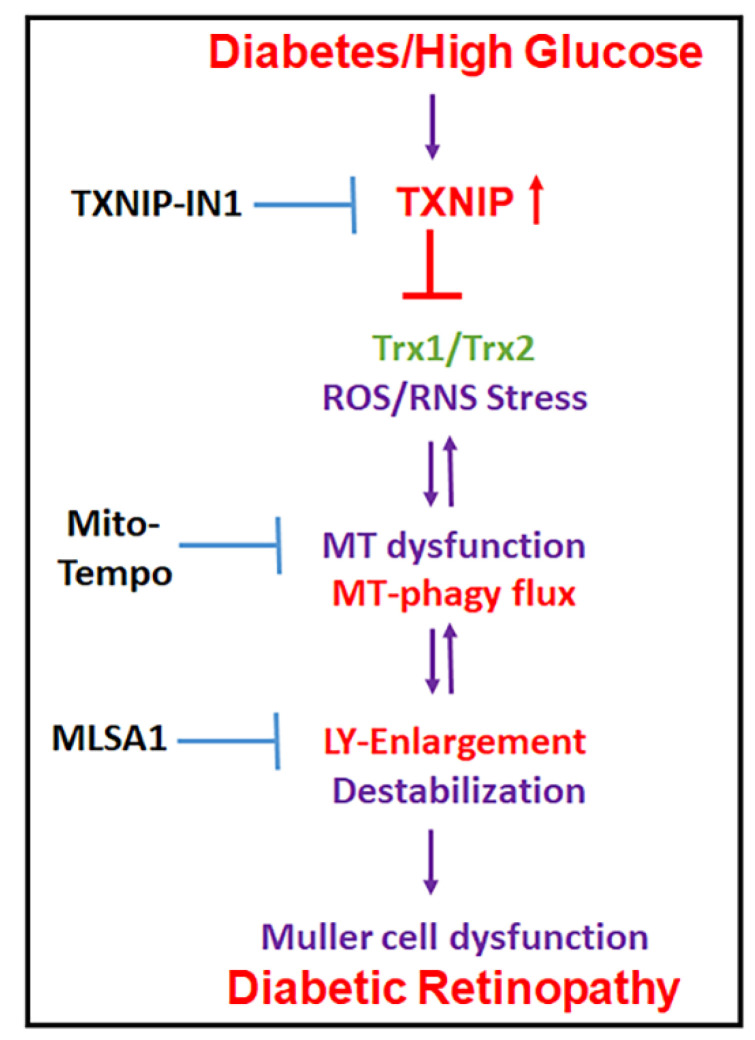
Summary of HG-induced TXNIP expression, redox stress, and the mitochondrial–lysosomal axis dysregulation in Müller cells. A potential therapeutic approach for targeting multiple steps at the cellular stress and organelle damage in the pathogenesis of DR. TXNIP-IN1, an inhibitor of the TXNIP-Trx interaction, Mito-Tempo, a mitochondrial matrix-specific superoxide scavenger, and ML-SA1, an agonist of the mucolipin transient receptor potential calcium channel 1 (MCOLN1/TRPML1), description in Discussion.

## Data Availability

Not applicable.

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
