# Peer review of "Potential Combination Drug Therapy to Prevent Redox Stress and Mitophagy Dysregulation in Retinal Müller Cells under High Glucose Conditions: Implications for Diabetic Retinopathy"

_diseases, 2021, doi:10.3390/diseases9040091_

Round 1

Reviewer 1 Report

The article entitled “Potential Combination Drug Therapy to Prevent Redox Stress and Mitophagy Dysregulation in Retinal Müller Cells under High Glucose Conditions: Implications for diabetic retinopathy” by Lalit Pukhrambam Singh and Takhellambam S. Devi reported that hyperglycemia via thioredoxin-interacting protein and reactive oxidative stress may participate in the development of diabetic retinopathy.

The article is of great scientific interest but a little difficult to read. The introduction may be shorter.

The authors developed molecular probes to study mitophagy in vitro and in animal models. They observed that diabetes induces TXNIP expression and redox stress just demonstrated by pictures (figure 2).

Figure 3: It will be more convincing if the authors present the individual values (figure3 D-E) instead of mean ± SEM, since they used the low limit of numbers for statistical analysis.

Figure 5: No statistical test can be used with n=3.

All the pictures are very beautiful and convincing.

Reviewer 2 Report

The Introduction: 

The introduction prepares the reader for the issues very well,  however, this section should be shortened. The introduction should briefly place the study in a broad context and highlight why it is important. At the moment, this part seems to be a review paper. 

Materials and Methods:

In my opinion, the first paragraph is not needed, the names of the company from which the materials have been purchased should be put during the description of the method. 

Similarly, the information about the volume of the reagents used during the protocols is not necessary. 

Results: 

All Figures and Tables should be inserted into the main text close to their first citation and must be numbered following their appearance. The results are well presented. 

Discussion:

Please consider the shortened Discussion. 

References:

References should be described according to the instruction.  In the text, reference numbers should be placed in square brackets [ ].

Reviewer 3 Report

Dear Authors,

The present study evaluates potential combination drug therapy to prevent redox stress and mitophagy dysregulation in retinal Müller cells under high glucose conditions. The presented studies will contribute to prevent retinal injury and disease progression in diabetes. And the research brings an excellent tool, Ad-CMV-2xmt8a-CG in the field.

Some changes of the manuscript should nevertheless be performed in order to improve its quality.

Following specific changes should thus be performed:

Minor changes

At line 69 is: …IL1b … .Should be: … IL1-b… .

At line 428 please delete the sentence: … lysosomal membrane permeabilization … .

At line 429 please delete the please delete the word ‘(LMP)’.

At line 187 is correctly: … 570–694 … , but at line 129 is … 4.5-5 …, and at references, for example 1 at line 498 is 44-84, should be … 4.5–5…, and 44–84, respectively. Similar errors are at other references. Recently, ‘em dash’ ( – ) longer than ‘en dash’ ( - ) is used between the numbers.Please correct carefully.

At lines 151–152 please add name of the paragraph 2 such “Materials”.

At lines 154–156 the country and region of customer are needed as in other cases, see line 161.

At line 203 is: …405 nm … . But at line 204 is: …488-nm… .

At line 69 is: …2000 rpm. for 5 min. … .Should be: … 2000 rpm for 5 min.… .

At line 519is …873-80…, should be … 873–880 …. Similar errors are at other references.

Major changes

In the Introduction, the aim of your study is not clear.

The language should be more concise.

Why did you not measure changes in inflammatory cytokine levels? 
